# Current Status and Consideration of Support/Care Robots for Stand-Up Motion

**Kensuke Nakamura [1,2] and Norihiko Saga [2,*]**

1 Rehabilitation Department, Amagasaki Medical Health Co-op Hospital, Amagasaki-shi, Hyogo-ken 661-0033, Japan; nakamura.kensuke@kwansei.ac.jp
2 Department of Human System Interaction, School of Science and Technology, Kwansei Gakuin University, Sanda-shi, Hyogo-ken 669-1337, Japan
* Correspondence: saga@kwansei.ac.jp; Tel.: +81-79-565-7042

**Abstract:** In order to make robots, which are expected to play an active role in the medical and nursing care fields in the future, more practical for use in rehabilitation, it is necessary to evaluate the current status of the design of these robots. Therefore, this paper aims to investigate the existing literature on standing motion assistance robots developed and reported to date and investigate each existing design technique from the perspectives of "Functions and Effects" and "Assist form and control." Then, we search and investigate papers written in English on standing motion assistance robots reported from 2008 to 2019 and organize the contents of the relevant papers into their different assistance modes and four categories related to design. As a result, the standing motion assistance robots are classified into three assist modes: partial assistance, total assistance, and both. The assistance forms are roughly divided into two types: a wearable type and a non-wearable type. It is also demonstrated that both the assistance forms adopt the same trends in terms of the control strategy design and system I/O relationships. On the other hand, power equipment tends to be different between the two forms.

**Keywords:** rehabilitation; support; care; robot; stand-up motion

## 1. Introduction

Japan is the most advanced country in terms of aging. As of 2015, the aging rate in developed countries was 26.6%, in Korea it was 13.1%, in the United States it was 14.8%, and in Sweden it was 19.9%. By 2060, it is estimated that about 1 out of 2.5 people in Japan will be aged 65 years or older [1]. Against this background, it is anticipated that an increase in the number of people requiring care and a shortage of human resources for caregivers will become more apparent in the future. The Ministry of Economy, Trade, and Industry (METI) and the Ministry of Health, Labour, and Welfare (MHLW) are concentrating their efforts on the field of caregiving and promoting efforts to develop and support the introduction of robotic care devices [2]. From these, an active push for the introduction of "robot" technology in the field of nursing, medical treatment, and welfare has been carried out in our country. On the other hand, Nakanishi [3] describes that there is a current situation in which rehabilitation robots have not yet come to practical use in the field from the viewpoint of a physiotherapist who belongs to the technology development headquarters of an enterprise. There are many issues that need to be examined concretely, such as factors that prevent practical application. One way to solve this situation is to consider what kind of robots have been introduced to date and to grasp what common problems and needs exist from the clinic side and the development side.

As of 2020, many robots have already been developed and reported both domestically and internationally, various problems concerning their practical application in the clinical field have been found, solutions have been proposed, and the verifications of these solutions have been carried out. Some of these have extended robots' use to aiding people in

standing up (sit-to-stand: STSs) and subsequent movements. Robots assisting with STS have been reported from various viewpoints, such as their mechanisms and control systems, in order to respond to social demands corresponding to each object illness, etc. However, robots such as Welwalk [4] and HAL® [5], which are increasingly being used at domestic rehabilitation sites and are beginning to be recognized, are but the tip of the iceberg, and a number of robots that are not yet widely known have been studied and reported.

This paper covers domestic and overseas previous research papers on STS assistance robots. STS is an essential movement for daily life and has clinical significance: STS is an essential movement for transitioning from sitting to standing posture, and higher STS movement ability has been reported to enhance the quality of life [6]. The clinical implications of STS are that daily STS can maintain hip and knee strength [7], that STS performance in the general elderly population and in people with who have suffered cerebrovascular accidents (CVAs) correlates with future mobility-related movements and fall rates [8,9], and that intensive STS training in people who have suffered CVAs is beneficial for learning mobility skills as well as increasing the loading rate on the paralyzed lower limb [10]. From the above, STS is a prerequisite movement for walking, has similarities with walking, and is a necessary movement for improving QOL.

As a clinical effect of these STS assistance robots, Tsukahara et al. [11] reported that a patient with complete paraplegia due to 10th-11th thoracic spinal cord injury was able to achieve STS with visual feedback and balance control through the use of HAL-5. Shiraishi et al. [12] reported that the use of a linearly driven STS training system in patients with after-effect of CVA increased the loading rate of the paralyzed leg during STS. These reports suggest that an STS assistance robot can compensate for a deficit or decrease in physical functions and may allow patients with STS difficulties to perform STS. The motions targeted by robots assisting lower limb functions can be broadly classified into those targeting STS only [13,14], those targeting STS and walking [15,16], and those targeting walking only [17]. In this paper, we focus on STS assistance robots, taking the former two patterns (STS only and STS and walking) as the assistance targets, and organize a literature survey by categorizing the elements that constitute each perspective to understand the current status of assistance robot design.

## 2. Methodology and Review Perspectives

### 2.1. Methodology

We conducted a systematic review in this study with the aim of using our findings to inform the development of orthostatic assistive devices. Publications were collected from the following databases:

- Google scholar,
- IEEE Xplore,
- PubMedCentral®.

These databases were selected based on their proceedings and their coverage of journals associated with robotics, the biomedical sector, life sciences, and their accessibility within the Kwansei Gakuin University library network.

We used the following search string to capture relevant papers:

("sit to stand" OR "standing up") AND ("assist" OR "support") AND ("robot" OR "device").

The search string was purposefully chosen to be generic in order to collect as many relevant papers as possible. The surveyed period was 12 years from 2008 to 2019. For the search, "Assist STS only" and "Assist both STS and walking" were retained from the titles and abstracts, while "Assist walking only" and other obvious non-target references were excluded. As a result, we obtained 87 articles published in international journals and the proceedings of IEEE international conferences. In addition, articles in which the proposed robot was presented in a report and in which the background of the development, the mechanism or control system, the evaluation, and the results were reported were selected as the final target. For the same device reported from different perspectives, the most recent

report year was aggregated as one. As a result of the above, the literature was narrowed down to a total of 45 articles for inclusion and evaluation.

*2.2. Review Methos*

2.2.1. Review Perspectives

We first describe the previous studies we investigated from two different perspectives: "Functions and Effects" and "Assist form and control." In the "Functions and Effects" section, assistance modes are classified into partial assistance and total assistance. In "Assist form and control," (1) assist system, (2) power unit, (3) control strategy, and (4) system are classified and organized. Based on the above, we will analyze the current status of STS assistance robots in the 45 papers surveyed.

2.2.2. Definition of Terms

The definitions of the terms used in this paper are described below.

- Partial assistance and total assistance:

Partial assistance is defined as supplementing a part of the force or torque required for operation, and total assistance is defined as supplementing the entire force or torque. When the two are not clearly distinguished, we use "assistance" as a general term.

- Power unit:

Refers to a device that generates force or torque that helps the robot to assist the wearer. Those that require an external power source are referred to as actuators (ACTs), while those that do not require an external power source are distinguished as devices (DEVs).

## 3. Results

The targets of assistive robots were broadly categorized as the elderly, people with motor disabilities, and central nervous system patients (spinal cord injury, cerebrovascular accident). The elderly include those with muscle weakness due to aging and those who need nursing care. There were a relatively larger number of research reports on robotics for the elderly, at 24 cases, including those overlapping with CVA diseases and excluding studies that did not specify the target population. On the other hand, there were 19 reports targeting CVA patients, including those overlapping with the elderly. There were 11 reports on patients with arthropathy or gait dysfunction for some reason, and most of them were targeted along with the elderly. These results are summarized in Figure 1. Since this article focuses on STS assistance robots, we will only discuss reports related to STS. Therefore, this paper does not cover reports that have been validated for effects other than STS, such as gait and control system simulation.

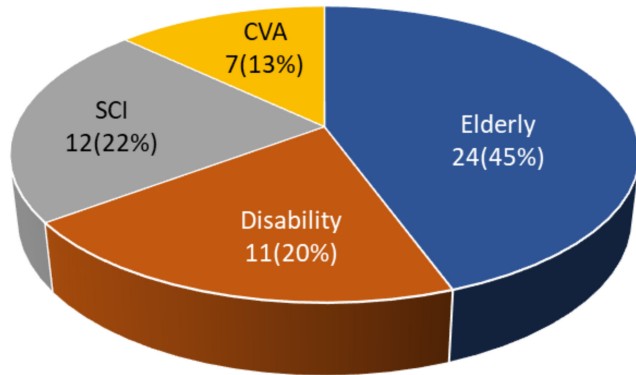

SCI: Spinal cord injury
CVA: Cerebro vascular accident

**Figure 1.** Summary of the classification of subjects.

*3.1. Functions and Effects*

The list of papers covered in this survey is shown in Tables 1 and 2. The table is roughly divided into wearable type and non-wearable type, which are described later in Section 3.2.1 Assist system. The functions of STS assistance robots can be broadly classified into three types of assistance modes: partial assistance, total assistance, and both. In the literature surveyed in this study, partial assistance was the most common, accounting for 31 out of 45 cases. On the other hand, eight of these were related to overall assistance, and only 3 of these were able to provide both. For the remaining three cases, no description of the assistance mode was found. The following is a description of the above three assistance modes, with examples of typical robots and their effects.

3.1.1. Partial Assistance

Twenty-four out of thirty-two cases of STS assistance robots providing partial assistance were designed for elderly people. In addition, those with lower limb motor disorders due to some problem, such as knee osteoarthritis, were the second most common target after the elderly. An example of partial support is Myosuit, which is discussed by Schmidt et al. [18]. The robot has a function to generate extension torque of the user's hip joint and knee joint through two-joint drive using an actuator and a cable. In addition, Myosuit places ligamentous rubber bands on the anterior thigh and posterior lower leg, and at 30% elongation it exerts about 215 N, which simulates the eccentric contraction of muscle. This gives the user about 26% of the knee peak torque during the sit-to-stand transition. Eto et al. [19] proposed an elbow-supporting device as an STS assistance robot with a shape that the elderly are familiar with in their daily lives. The robot is placed at the side of the user, and when the user places his or her arm against it and applies weight, the armrest moves horizontally and vertically to assist the user in STS. This applies 40–60% of the body weight to the armrest and significantly reduces the user's rectus femoris muscle activity during STS.

**Table 1.** Assist functions and effects (Wearable type).

| Year | First Author | Assist Level | Function | Effect | Reference |
|------|-------------|-------------|----------|--------|-----------|
| 2011 | Tsukahara | T | Estimating STS start intention by preparatory movement. Control COP within reference range during STS. | The operating COP trajectory was controlled to 40.1% of the reference range. | [11] |
| 2016 | Kamali | P | Knee joint flexion/extension torque assistance using a linear actuator. | Knee joint work was significantly reduced. | [14] |
| 2013 | Tanabe | T | Use with custom walker. Start STS total-assist with the signal to lean forward in the trunk. | ND | [15] |
| 2017 | Schmidt | P | Efferent/Afferent torque assistance for hip and knee joints with actuator connection cables and rubber bands. | 26% peak torque was provided to the knee joint. Gluteus maximus activity reduced by about 60%. | [18] |
| 2015 | Kozlowski | T | Use with a front wheel walker. Attach the exoskeleton to the ceiling rail tether to assist STS. | One-third of subjects acquired STS with less training than physical assistance. | [20] |
| 2019 | Urendes | P/T | STS assist with suspension harness. | ND | [21] |
| 2014 | Hwang | P/T | Integrated structure of exoskeleton, electric wheelchair and lift. Weight assist during STS. | STS completed in 31 s on average. | [22] |

**Table 1.** *Cont.*

| Year | First Author | Assist Level | Function | Effect | Reference |
|------|-------------|-------------|----------|--------|-----------|
| 2012 | Huo | P | Joint torque estimation by torque observer. Hip and knee joint torque assistance. | Maximum muscle activity of knee extension decreased by an average of about 12%. | [23] |
| 2011 | MORI | T | Use with Lofstrand Crutch. Full STS assist with no backdrive. | STS completed in 5.0 to 11.5 s. | [24] |
| 2014 | Kimura | P | Assists hip and knee joint torque by estimating motion intention. | Rectus femoris activity decreased to 51.7%. | [25] |
| 2012 | Quintero | T | Estimating STS start intention by COP displacement Guidance with Total assist for standing, walking, and sitting | Operation completed in 114 s on average Consistency between enforcement | [26] |
| 2018 | Önen | T | Use with crutches. Hip and knee joint torque assistance. | ND | [27] |
| 2019 | Vantilt | P | Torque assistance for 3 joints of lower limbs. | The device had the required DOF and ROM. | [28] |
| 2016 | Park | ND | Torque assistance for 3 joints of the lower limbs with 10 actuators. | ND | [29] |
| 2018 | Wu | T | Combined with clutch with remote control Total assist for hip and knee joint torque | STS speed is twice as fast as KAFO. | [30] |
| 2016 | Asselin | T | Use with crutches. Total assist in hip and knee power. | ND | [31] |
| 2017 | Chen | T | Use with smart clutch. Assists hip and knee torque. | Sufficient joint torque was supplied for STS. | [32] |
| 2019 | Zhu | P | Assist knee joint torque with a custom manufactured motor. | ND | [33] |
| 2010 | Eguchi | P | Changes in trunk angle cause STS assistance by the device. | Quadriceps activity decreased by 30-50% of maximum muscle activity. | [34] |
| 2013 | Mefoued | ND | Allows robust control over disturbances. Knee joint torque assist during STS. | Robustness of control against disturbances demonstrated. | [35] |
| 2014 | Olivier | P | Joint design that emulates the DOF of the hip joint. Hip torque assistance. | The required hip joint torque could be fully exerted at 70° hip flexion. | [36] |
| 2018 | Junius | P | Hip flexion/extension torque assistance with a device equipped with redundant joints. | Reduced muscle activity in the gluteus maximus and biceps femoris. Oxygen consumption decreased. | [37] |
| 2018 | Wang | ND | Exercise intention recognition by BCI. Hip and knee joint torque assistance. | Visual image was about 10% higher in recognition accuracy. | [38] |

ND: No description; P: Partial assistance; T: Total assistance.

**Table 2.** Assist functions and effects (Non-wearable type).

| Year | First Author | Assist System | Assist Level | Function | Effect | Reference |
|---|---|---|---|---|---|---|
| 2013 | Shiraishi | C | P | Use visual feedback to reduce the difference in utilization between both legs. | Increased floor reaction force on the affected lower limb. | [12] |
| 2012 | CAO | S | P | Motion estimation with two ropes. STS trajectory guidance and reduction of lower limb burden. | Floor reaction force decreased compared to self-movement. | [13] |
| 2018 | Takeda | L/W | P | Estimate motion with the minimum number of sensors to reduce the burden on the lower limbs. | The estimated time of movement was 0.005 s. The average estimation error was 0.145 s. | [16] |
| 2015 | Eto | H | P | Lateral mobile armrests reduce the burden on the lower limbs. | Rectus femoris activity decreased significantly. | [19] |
| 2015 | Hoang | L/W | P | Reduction of lower limb burden by manipulator that draws the optimum trajectory. | It was judged that the optimum trajectory can be realized by the force of the actuator. | [39] |
| 2015 | Tsusaka | H | P | Imitate professional assistance skills. Posture guidance in the horizontal direction and lower limb burden. assistance in the vertical direction | It was verified that relatively appropriate assistance is possible under hybrid control. | [40] |
| 2012 | Salah | L/W | P | Imitate professional assistance skills. STS assistance coordinated with user posture estimation. | The maximum error of the estimate was about 0.04 m. | [41] |
| 2015 | Asker | L/W | P | Reduction of lower limb burden by 3DOF manipulator. | The maximum power of the power unit is 63% of the body weight. | [42] |
| 2011 | Saint-Bauzel | L/W | P | STS assistance with 2DOF manipulator. | A steering wheel trajectory that does not cause discomfort to the user was guided. | [43] |
| 2013 | Yuk | L/W | P | STS assistance by changing the angle and height of the armrest. | ND | [44] |
| 2010 | Carrera | S | P | Robot towed by rails placed on the ceiling. STS assist by prism joint. | ND | [45] |
| 2010 | NANGO | C | P | Seat-off assist by the seat follows the thigh angle. | The generated floor reaction force has decreased. | [46] |
| 2012 | Morita | L/W | P | Emulate professional assistance skills. Manipulator reduces STS lower limb burden. | Knee joint load reduced by 0.5 Nm/kg. | [47] |

**Table 2.** *Cont*.

| Year | First Author | Assist System | Assist Level | Function | Effect | Reference |
|------|--------------|---------------|--------------|----------|--------|-----------|
| 2012 | Bulea | L/W | P | Combined with functional electrical stimulation. STS assistance that does not require the user's upper limb muscle strength. | Floor reaction force was significantly reduced. | [48] |
| 2013 | Matjacic | C | P | Natural STS guide with a folding chair-type device. | Floor reaction force and muscle activity decreased. High similarity of movement patterns was observed. | [49] |
| 2016 | Fraiszudeen | C | P | Seat-off assistance with pneumatic actuators. | Raised the seat to 45 degrees within 10 s with a force of 200 N. | [50] |
| 2017 | Dong | L/W | P | Assisting the natural STS trajectory. | ND | [51] |
| 2018 | Sogo | C | P | Use only passive actuator. Seat-off assistance with gas springs. | Maximum hip and knee torque reduced significantly during seat-off. | [52] |
| 2011 | Bae | C | P | Adjust the height and angle of the seat to make seat-off easier. | The higher the seating surface, the less rectus femoris activity. Rectus femoris activity is significantly reduced at a seat angle of 15 degrees. | [53] |
| 2017 | Suzuki | C | P | STS assistance by constant speed seat rotation. Seat-off assistance reduces the burden on the lower limbs. | Maximum floor reaction force significantly reduced. | [54] |
| 2014 | Lu | C | P | STS motion estimation by COP pattern recognition. Seat-off assistance by changing the angle and height of the seat surface during STS. | ND | [55] |
| 2011 | An | H | P | Handrail that moves horizontally/vertically in coordination with the lower limb joint angle. | ND | [56] |

ND: No description; P: Partial assistance; T: Total assistance; C: Chair; S: Sling; LW: Leaning-on/Walker; H: Handle.

### 3.1.2. Total Assistance

STS assistance robots that provide total assistance were mostly reported to be listed for SCI patients, with 6 out of 8 reports. Those providing holistic assistance were characterized by a high percentage of exoskeletal robots and the use of a clutch or walker. An example of holistic assistance is Ekso by Kozlowski et al. [20]. This powered exoskeleton robot is intended to help SCI patients achieve motor learning effects for STS and gait. The robot has an exoskeleton attached to a ceiling rail tether and is used with a walker or clutch.

Using the Ekso, patients were able to learn STS in 5 (5.4–10.6) sessions, compared to 18 (8.3–27.7) sessions when practicing learning STS while touching the body. On the other hand, during this survey period no non-wearable total assistance robots were found. In a report prior to 2008, for example, the stand-up robot support device reported by Kamnik et al. [57] was a non-exoskeleton STS assistance robot that provided total assistance. In this

paper, the patient first uses the robot in the sitting position and then shifts to the standing position by extending the lower limbs while keeping the buttocks in contact with the seating surface. The robot lifts 99% of its body weight and reduces the strain on the lower limbs.

### 3.1.3. Partial and Total Assistance

For STS assistance robots that can choose between partial and total assistance, we first describe HYBRID by Urendes et al. [21]. This robot features a powered exoskeleton (H1) and a powered walker (REMOVI) with a powered sling arm. H1 is connected to REMOVI by a harness, and REMOVI can provide partial or total assistance when the user stands up from a wheelchair or other device. H1 is connected to REMOVI by a harness and REMOVI can provide partial or total assistance when the user stands up from a wheelchair or other device. Another example is the BWS training mobile exoskeleton system of Hwang et al. [22]. This robot is designed for the rehabilitation of SCI and CVA patients and consists of a powered exoskeleton integrated with a power wheelchair and a power lift. The lift part can control the weight bearing of the user. The test subjects using this robot were able to shift from a sitting position to a standing position in an average of about 31 s. After standing, the subject can be trained to walk or climb stairs by partial or full unloading.

### 3.2. Assist Form and Control

The contents of the assistance and control in the literature surveyed in this study are summarized in Table 3. In this section, we categorize and organize them into (1) assistance modes, (2) power equipment, (3) control strategy, and (4) systems. The assistance mode classifies the means by which each assisting robot assists the user's STS, while the power equipment classifies the actuators and devices with which the robot assists the user's STS. In control strategy, we classify whether the control law implemented in the robot is position control or force control, and in system we classify the input–output relationship of the controller signals as single-input single-output or multi-input multi-output.

**Table 3.** Summary of the assistance forms and control of the survey.

| Year | First Author | Type | ACT | Control Strategy | System | Reference |
|------|------|------|------|------|------|------|
| 2010 | Tsukahara | W | ACT Spring | Hybrid | MIMO | [11] |
| 2016 | Shiraishi | NW | L-ACT | Position | SISO | [12] |
| 2012 | CAO | NW | Motor | Impedance | SISO | [13] |
| 2016 | Kamali | W | Motor Spring | Impedance | SISO | [14] |
| 2018 | Tanabe | W | Motor | ND | ND | [15] |
| 2018 | Takeda | NW | L-ACT | ND | ND | [16] |
| 2017 | Schmidt | W | Motor Rubber | Force | SISO | [18] |
| 2015 | Eto | NW | ACT | ND | ND | [19] |
| 2015 | Kozlowski | W | Motor | ND | ND | [20] |
| 2019 | Urendes | W | Motor | Position | ND | [21] |
| 2013 | Hwang | W | Motor Spring | Position | SISO | [22] |
| 2016 | Huo | W | Motor Spring | Force | SISO | [23] |
| 2011 | MORI | W | Motor | Position | SISO | [24] |
| 2014 | Kimura | W | Motor | Force | SISO | [25] |

**Table 3.** *Cont.*

| Year | First Author | Type | ACT | Control Strategy | System | Reference |
|------|-------------|------|-----|------------------|--------|-----------|
| 2018 | Quintero | W | Motor | Position | ND | [26] |
| 2014 | Önen | W | Motor | Position | MIMO | [27] |
| 2019 | Vantilt | W | Motor Spring | P/F | SISO | [28] |
| 2015 | Park | W | Motor | Position | SISO | [29] |
| 2018 | Wu | W | Motor | ND | ND | [30] |
| 2016 | Asselin | W | Motor Spring | ND | ND | [31] |
| 2017 | Chen | W | Motor | Position | SISO | [32] |
| 2019 | Zhu | W | Motor | Force | SISO | [33] |
| 2018 | Eguchi | W | G-spring | Less | Less | [34] |
| 2012 | Mefoued | W | Motor | Position | MIMO | [35] |
| 2014 | Olivier | W | L-ACT | Force | SISO | [36] |
| 2018 | Junius | W | Motor Spring | ND | ND | [37] |
| 2018 | Wang | W | Motor Spring | ND | ND | [38] |
| 2015 | Hoang | NW | L-ACT | Position | SISO | [39] |
| 2015 | Tsusaka | NW | Motor | P/F | SISO | [40] |
| 2012 | Salah | NW | Motor | Position | MIMO | [41] |
| 2015 | Asker | NW | ACT | Position | SISO | [42] |
| 2011 | Saint-Bauzel | NW | H-ACT | P/F | SISO | [43] |
| 2013 | Yuk | NW | L-ACT | ND | ND | [44] |
| 2011 | Carrera | NW | Motor | ND | ND | [45] |
| 2010 | NANGO | NW | Mechanism | Less | Less | [46] |
| 2012 | Morita | NW | Motor | P/F | SISO | [47] |
| 2012 | Bulea | NW | G-spring | Less | Less | [48] |
| 2016 | Matjacic | NW | Motor | Position | SISO | [49] |
| 2016 | Fraiszud-een | NW | P-ACT | Position | SISO | [50] |
| 2017 | Dong | NW | L-ACT | Position | SISO | [51] |
| 2018 | Sogo | NW | G-spring | Less | Less | [52] |
| 2011 | Bae | NW | L-ACT G-spring | ON/OFF | SISO | [53] |
| 2017 | Suzuki | NW | Motor | ON/OFF | SISO | [54] |
| 2014 | Lu | NW | L-ACT | Position | SISO | [55] |
| 2011 | An | NW | L-ACT | Position | SISO | [56] |

Type—W: Wearable, NW: Non-Wearable; ACT—L-ACT: Linear actuator, H-ACT: Hydraulic actuator; Control strategy—P/F: Position and force control switching, Less: No control, ND: No description.

### 3.2.1. Assistance System

The assistance system of robots for STS and walking motion is roughly classified into two types—"wearable" or "not wearable" by the user—depending on the type of robot

used and the support motion. In addition, STS assistance systems of the non-wearable type can be mainly subdivided into four types (Figure 2). The chair type [58] has a raised or inclined seating surface to convey the force to the buttocks of the user. In the sling type [59], a manipulator raises the trunk through a cable or a belt. In the leaning-on type (including walker type) [60,61], part of the body weight is rested on the support arm, and the support arm lifts the body. The handle type [62] reduces the load on the lower limb when the user grasps the handrail provided on a toilet bowl, chair, or bed, partially bearing their weight. The breakdown of these assistance methods is summarized in Figure 3.

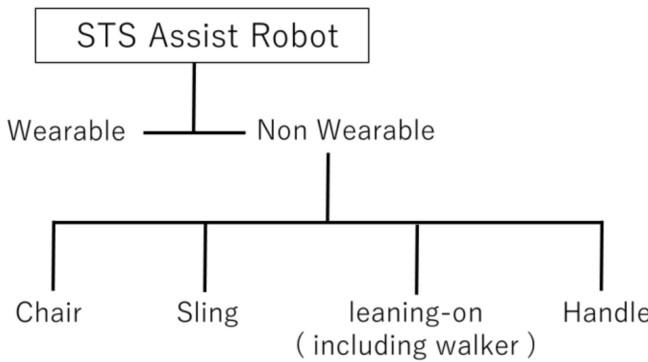

**Figure 2.** Categorization of sit-to-stand (STS) assistance robots.

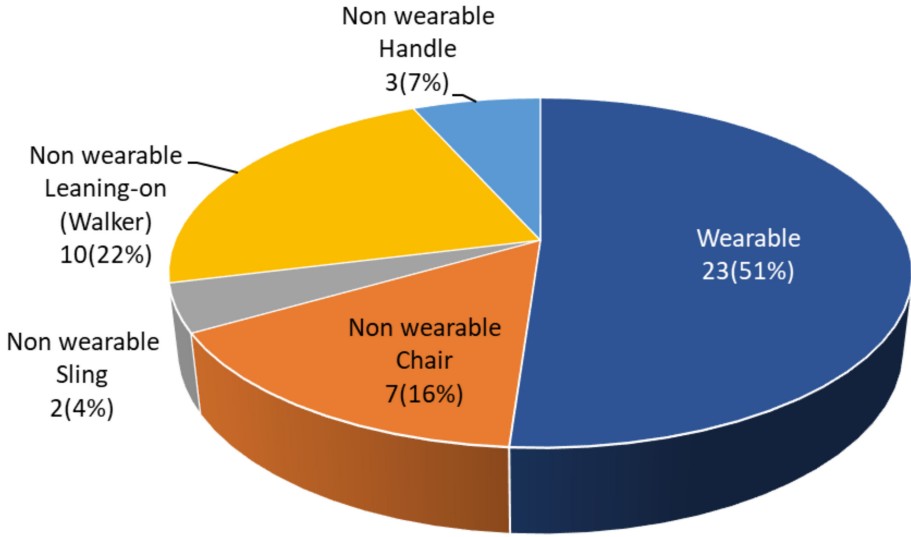

**Figure 3.** Summary of assistance methods.

An example of assistance in a wearable configuration is "E-ROWA" from Huo et al. [23]. E-ROWA is a lower extremity skeletal robot with controllers, drivers, and batteries on its back and four actuators at the hip and knee joints. It has a control system that estimates the torque generated using an encoder at the hip joint and knee joint of the wearer and a floor reaction force sensor at the shoe sole without using an expensive force/torque sensor or an electromyogram sensor, which are easily influenced by external factors, and it can partially provide the joint torque required for STS by a wearer. On the other hand, MORI et al.'s [24] exoskeleton-type wearable robotic "ABLE" is exemplified as an example of support. ABLE is for people with spinal cord injuries. It has ACT in the hip joint and the knee joint, and the author mentions that the attitude control is made possible by using a telescopic loft strand stick mounted on the robot control switch. The actuator of the joint does not cause back drive, the joint angle of the robot is maintained even in the power OFF state, and the

total joint torque necessary for STS can be provided to the wearer. In the standing motion, it shifts to the standing posture by the lower limb joint torque provided, while the force of the upper limb is used with the loft strand stick.

Examples of assistance of the non-wearable type include the double-rope system of CAO et al. [13]. This equipment uses a sling with two ropes each at the front and rear connected to the wearer. The wearer is equipped with a connecting jacket as a harness, and the tension of two ropes in the front and rear is transmitted to the wearer through a jacket. Each rope is tension measured in real time by the load cell, the servomotor of the rear rope determines the assist force, and the front rope induces the locus of the center of gravity of the standing motion. This is a chair-type equipment in a seesaw shape in which hydraulic ACT is provided in a mechanism with three degrees of freedom. It has a rotary motion segment and a translational motion segment, and a seating surface is raised at a horizontal angle by a leader–follower system with a rotary motion segment. A lift support of 99% is possible, and the standing action can be carried out by the combined use of the force of the upper limb and functional electric stimulation for the lower limb, even in a completely paraplegic patient.

### 3.2.2. Power Unit

In assisting with STS, some or all of the required lower limb joint torque should be provided to the user by the robot. Power devices that convert power sources such as electricity into mechanical energy such as joint torque in robots are diverse, with different mechanisms and solutions used by the robots, and the power sources are also different. Reviews on power equipment are summarized in Figures 4 and 5. In the literature surveyed in this paper, as the device for generating joint torque in a robot, those using only ACTs comprised 29 cases out of 45, those using only DEVs comprised 4 cases, those using both together comprised 9 cases, and a mechanism using the user's own power comprised 1 case. Thirteen of the 29 cases using only ACTs were wearable. Of these, 12 were electric motors and 1 was an electric linear ACT. On the other hand, 16 out of 29 cases using only ACTs were of the non-wearable type. Of these, seven were electric motors, seven were electric linear ACTs, one was a hydraulic linear ACT using hydraulic power as a power source, and one was a bellows-type pneumatic ACT. Motor and linear ACTs that convert electricity to mechanical energy from the above were often adopted, and the ratio of linear ACTs was higher in the non-wearable type. In addition, one was of the wearable type and three were of the non-wearable type using only DEVs, and three cases out of four were gas springs. In addition, 9 of the 10 cases of the combined use of ACTs and DEVs were of the wearable type, 8 cases used a combination of motor and spring, and 1 case used a combination of some kind of ACT and a spring. There was one case of the non-wearable type using ACTs and DEVs, together with one that used a linear ACT and a gas spring.

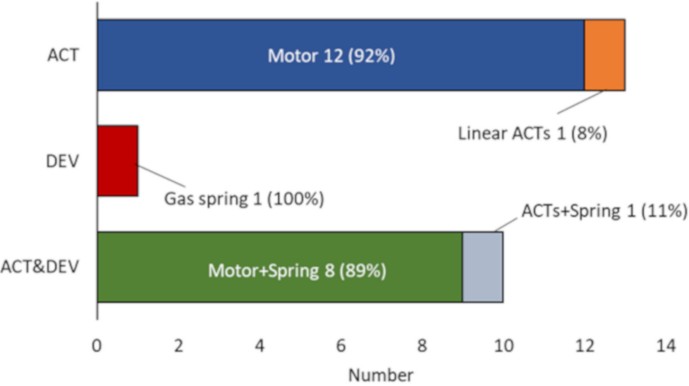

**Figure 4.** Summary of power unit classification in wearable robots.

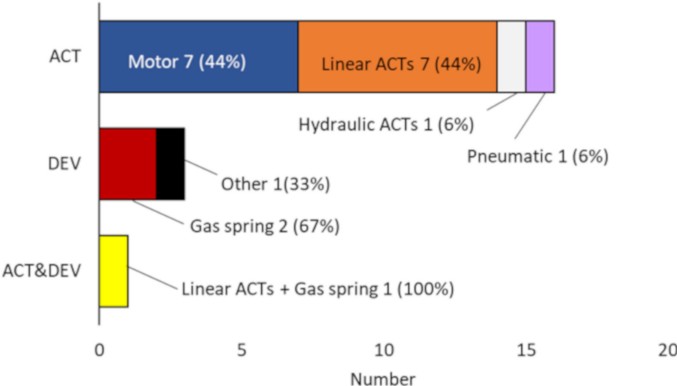

**Figure 5.** Summary of power unit classification in non-wearable robots.

When only ACTs are used in the wearable type, Tanabe et al's "WPAL" is the most common report on the use of motors [15]. WPAL is a robot targeting bilateral lower limb paralytic patients with spinal cord injury. The hip joint of the robot is designed in the inside position as in the conventional Primewalk, and it is a lower limb external skeleton robot with combined use with a wheelchair in mind. WPAL is equipped with electric motors with identical specifications in the three joints of hip, knee, and foot, and these generate standing/sitting motion and walking motion patterns controlled in real time by CPUs with a handy switching operation using a combined use walker.

Examples of combinations of ACTs and DEVs that have been reported to be secondary to the use of ACTs alone include "FUM-Knee Exo" by Kamali et al. [14]. This robot is equipped with a linear-series elastic actuator (LSEA); the rotational motion of the motor becomes the linear motion of the ball screw, and it is transmitted as an output through a spring set. This LSEA enables the precise and robust control of human–robot interaction. Linear ACTs, which are most widely chosen along with motors in non-mounted ACTs, are devices that convert the rotational motion of motors into linear motion and are used in the "MOBOT" of Hoang et al. [39]. MOBOT causes linear ACTs to exert a force on the links in a linear-dynamic direction, creating torque in the robot arm joint, and the end-effector provides a portion of the lower limb joint torque required for to help the user with STS. In DEVs alone, gas-pulling has been the most chosen method, including STSs and the mobile-assisted robots using mobility beagles reported by Tsukahara et al. The robot has an external skeleton to be worn on the lower limb on a mobile platform, and a total of four gas springs provide users with lower thigh anteversion assist torque and knee extension assist torque in STS.

3.2.3. Designing of Control Strategy

A total of 39 out of 45 robots used ACTs in this study. Among them, there were 31 cases in which the control strategy was specified, and the remaining 8 cases were not specified. Of the 31 control strategy design, 5 cases used force control, 17 cases used position control, 1 case used hybrid control, and 2 cases used impedance control. In addition, there were 4 switchable cases of position control and force control (P/F), 2 cases of ON/OFF control, 10 cases of no description, and 4 cases of them having no control. The following are classified into force control, position control, and compliant motion control hybrid control and impedance control. The results are summarized in Figure 6, and the previous studies using the respective control strategy are mentioned.

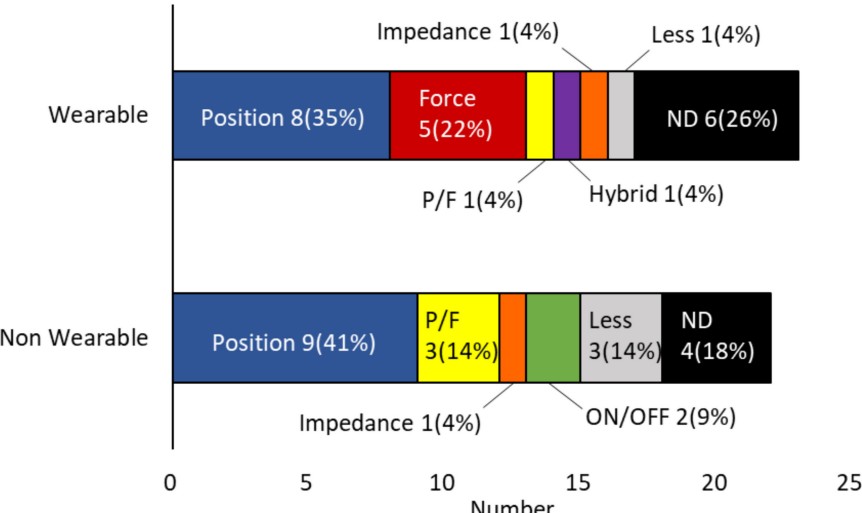

**Figure 6.** Summary of control strategy classification.

Force Control

One example of the force-control robots used in three of the wearable types is the report of Kimura et al. [25] which used "TTI-Exo." TTI-Exo is an electric whole-body exoskeleton robot that estimates a wearer's attitude transition from an input signal detected by an angle sensor installed at the joint of the robot. The output of the ACT designed for the hip joint and knee joint of the robot is controlled so that the assisting torque suitable for the transition can be obtained. The Myosuit reported by Schmidt et al. [18] is a textile robot that uses both tendon actuators and rubber bands to assist the muscle torque of the user's hip and knee joints. The robot calculates the knee joint angle from the trunk and lower leg angles monitored by the IMU and sends a feedback signal to the PID controller along with the tendon cable length. The exerted torque of the tendon actuator is controlled in real time.

No reports were found to be designed only with force control in the non-wearable type. The robots using force control adopted hybrid control or impedance control, which is described later, and were seen when the switching between force control and position control was made possible by the assisting phase or the user's arbitrary setting. The linear stage system of Tsusaka et al. [40] is mentioned as a report of switching between force control and position control using an assist phase. This equipment classifies the posture transition in the standing operation into four phases, and the system is designed to use position control in the phase which inclines the upper body forward and force control in the phase which floats the buttocks and extends the lower limbs; the adjustment of the assistance force suitable for the user's lower limb muscular force was realized under the force control.

Position Control

An exemplary position-controlled robot used in eight of the wearable types is the "Vanderbilt powered lower limb prostheses" from Quintero et al. [26]. The posture of the orthotic is estimated from the angle measurement based on the Hall effect in each knee joint and hip joint, a triaxial accelerometer, and a gyroscope in each thigh. Using the estimated values of this posture, four motors give the torque of sagittal plane to the knee joint and hip joint to control the position (in this case, joint angle). The non-wearable type uses position control in 9 cases, one of them is an "E-JUST Assistive Device" reported by Salah et al. [41]. The robot estimates the user's posture through inputs from the user's lower limb and an inertial sensor worn on the fuselage; it controls the trajectory of the end-effector to imitate the real motion of the caregiver. The operation of the end effector is performed by a motor located at the joint part of the robot.

Hybrid Control/Impedance Control

In the literature investigated this time, hybrid control is used in the "HAL" of Tsukahara et al. [11], which is a wearable type. This robot measures the relative angle by attaching a potentiometer to each joint of the lower leg, arranges a three-axis accelerometer in the control box to be attached to the waist, measures the absolute angle of the fuselage, and calculates the center of gravity by attaching a pressure sensor to the foot sole of the shoe. They recognize the wearer's operation intention by using the center of gravity position and floor reaction force sensors, and give assistance/assistance torque to each joint using the power unit in each joint. In this control, each torque and joint angle of the ankle joint, the knee joint, and the hip joint are controlled simultaneously.

Examples of impedance control include FUM-Knee EXO of Kamali et al. [14]. This equipment measures the deflection of the spring with a porcelain linear encoder; the power of LSEA is used for the estimation, the rotational angle of the ankle joint division is measured by the encoder, and it is used for the track estimation. Four force sensors are embedded in the foot to estimate the power of LSEA inversely dynamically and to detect the seat-off of the buttocks. The controller of this machine is composed of a force control feedback loop and an impedance control loop, and the relation between the LSEA power and the exoskeleton and the joint angle of the user is controlled.

### 3.2.4. System

Regarding the input–output relation of the system, SISO, which has been used for classical control theory since around 1930–1940, has been developed from around 1950–1960 to the present MIMO, which is represented by modern control theory [63]. Regarding the relation between the signal input and output of a system in the literature surveyed this time, there were 25 cases in which SISO was selected, 4 cases in which MIMO was selected. In the 43 cases excluding 4 cases without ACTs, 12 cases did not specify the system in each report. We summarize the results in Figure 7.

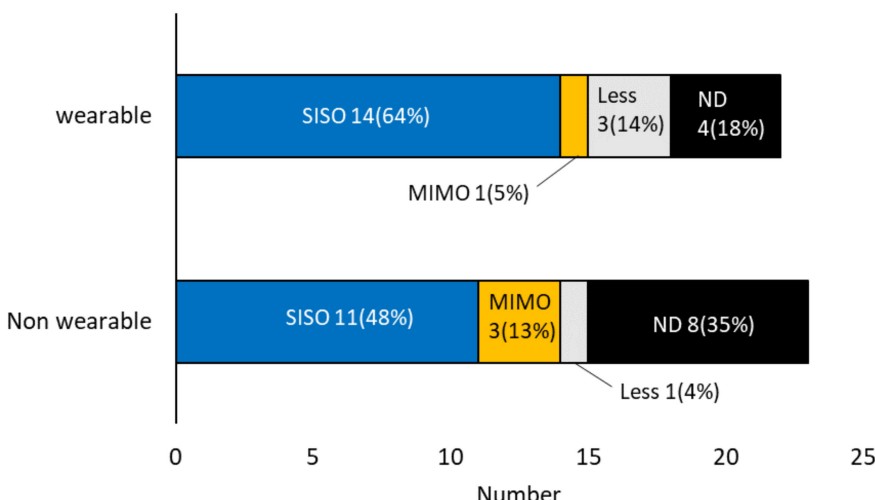

**Figure 7.** Summary of the system classification.

One example of SISO includes the multi-functional locomotion aid of Asker et al. [42]. This equipment was equipped with a plane parallel manipulator on an active walker, and not only STS but also movement and standing position maintenance were developed as support object operations. The manipulator is controlled by the PD controller, and the output signal is the amount of change in the linear motion ACT length that drives the manipulator, and the input signal is the voltage that is obtained by comparing the output signal (including disturbance) and the ACT length. Therefore, the SISO system controls the attitude of the manipulator.

One example of MIMO is a report of a walking assistance exoskeleton (WSE) by Önen et al. [27], which is a wearable robot with four motors driving both joints, allowing movement in the sagittal plane of the hip and knee joints. The WSE uses elements of both neural networks and fuzzy control. The control method is based on a control method to optimize the control function. The input is the joint angle error and the joint speed error between the target value and the current value of the robot-driven joints; the control voltage is output to each ACT from a controller with four different control functions to control the motion of the robot to the wearer.

## 4. Consideration

### 4.1. Functions and Effects

#### 4.1.1. Partial Assistance

In this survey, STS assitance robots that provide partial assistance were most frequently reported, and the typical target population was the elderly. As the elderly age, their lower limb muscle strength declines, making it difficult for them to perform STS and walk, which are essential for getting them to their destinations in daily life. The importance of preventing lower limb muscle weakness in the elderly is already known, and a method to prevent lower limb muscle weakness while maintaining activity level is to assist movement and utilize residual muscle strength. In the STS assistance robots with the purpose of utilizing the residual muscle force, seat-off from the seat surface, weight support by a manipulator, and torque support of the hip joint and knee joint are carried out. In the effect verification, dynamic evaluations of the floor reaction force and decreased effect of muscle and leg activity are mainly carried out. In fact, the elderly have difficulty in generating the floor reaction force required at the time of leaving their seat because of the insufficient exertion of the joint torque caused by the decrease in the leg muscle strength with aging. For the elderly, it seems to be effective to supplement a part of the joint torque with the assistance robot, which provides partial support, and to require the user to lift themselves otherwise.

#### 4.1.2. Total Assistance

Total assistance robots tended to be wearable and targeted at SCI patients with lower limb paralysis. STS assistance robots with total assistance are characterized by the fact that they often use a clutch or walker in addition to a powered exoskeleton. Total assistance robots provide all the joint torque required for STS, except in cases such as HAL® by Tsukahara et al., but they do not have the capability of postural control to stabilize the user's balance during STS or walking. The target of the total assistance robots investigated in this study is SCI patients with bilateral lower limb paralysis, such as those with parietal injuries who have residual upper limb function. Therefore, it is expected that most of the users' standing balance will be secured by using a clutch or walker with their remaining upper limbs. The user of the total assistance robots is paralyzed in both lower limbs, and therefore cannot initiate standing movements by exerting the power of the lower limbs. Therefore, the clutch to be used together is equipped with a switch for starting STS, or the intention estimation of STS movement by posture change is implemented. Therefore, the effectiveness of the system is often verified by the kinematic verification of practical aspects, such as whether STS can be performed as intended by the user and the speed of the STS operation.

#### 4.1.3. Partial and Total Assistance

Although the number of cases related to STS assistance robots with the option of partial or total assistance was small in the literature surveyed in this study, one of the common features of these robots is that they are targeted at rehabilitation patients. Rehabilitation patients include the elderly and patients with musculoskeletal disorders and central nervous system disorders. The elderly and patients with musculoskeletal disorders can choose partial support, while patients with central nervous system disorders such

as SCI and CVA can choose partial or total assist, depending on the degree of paralysis of the patient, and are considered to have both the partial and total support functions as mentioned above.

### 4.2. Assist Form and Control

#### 4.2.1. Assist System

The robots proposed in the literature as a survey object can be largely classified into a "wearable type" and a "non-wearable type," and the non-wearable type can be classified into four types of modes: "chair type," "sling type," "leaning-on type," and "handrail type."

In the background in which the assist system by wearable/non-wearable differs, it is described that the wearable type can be expected to be versatile in various everyday situations for the non-wearable type. Furthermore, regarding the number of studies related to each assistance method investigated, it seems that the number of wearable types targeting spinal injury and motor function impairment (such as deformable arthrosis), besides elderly people, tend to be higher than the number of non-wearable types. Individuals with spinal cord injury have greatly reduced lower limb muscle function [64], and motor dysfunction caused by aging, such as knee osteoarthritis which is also known to significantly reduce the lower limb muscle function [65]. Since the wearable type can provide a large assistance force to the user by a link which is parallel to the actuator and body segment adjacent to the joint, it seems to be chosen for spinal cord damage and motor handicap as a specification in which support and high-level assistance become possible.

On the other hand, the non-wearable type is relatively large in scale and has many functions for movement assistance in addition to standing [41,42,66]. It is considered that it is assumed that it will be used in a limited environment and indoor situations, such as nursing care facilities. Therefore, it is considered that many elderly users are selected as the main target users, mainly for use in buildings where assistance is provided on a daily basis. In the case of elderly users, the necessity of preventing further increase in the amount of nursing by drawing out the remaining function of the person is described in many previous studies, so that the non-wearable type tends to use the assist system as assistance. A handrail type in which a chair type is installed is used when an assist is necessary, while only a standing operation is possible by itself, and a handrail type of a leaning-on type or a pedestrian type is used, when an assist is necessary for a remaining function even in moving itself. While the leaning-on type can deposit a part of its own weight in the equipment, on the other hand, the handrail type requires the remaining function of the lower limb and the upper limb force to lift own weight, so that the leaning-on type can obtain more large assist force than handle type. However, the handle type also has advantages. It has been reported to be effective for subjects who cannot maintain balance during standing movement due to ankle dysfunction [67].The sling type is capable of leaving the weight on the device more than the leaning-on type, and it seems to be the equipment for the user who needs the assistance closest among the non-wearable types.

#### 4.2.2. Power Unit

Robots assist the movement of the body through interfaces; either the power source is converted to mechanical energy by ACTs or the dead weight is utilized for joint drive by DEVs. Based on the results of this investigation, the motor was selected in both cases of ACTs only and the combined use of ACTs and DEV in the wearable type. The reason why the motor is adopted in the wearable type is that the wearable type is light and compact and the wearable feeling is good [28]. Linear ACTs tends to take a large moment arm, while the size of the driving device increases because the device is placed away from the human body joint. Although a wearable type [68] has also been reported in which a motor is placed at a location away from the joint and an articulated joint is driven by fewer motors via a cable, in many cases individual drive units have been miniaturized by placing the motor adjacent to the human body joint. In addition to reducing the size and energy consumption

of the joint drive system by combining it with a spring or other DEVs, some researchers have used DEVs such as springs together for joints that do not require control to ensure the joint flexibility of the wearer [69].

On the other hand, many non-wearable-type robots are of the chair type, which is installed in a fixed position, and the hand-held type, which is a walker type which can move the main body and which leaves the body to the support arm. These work by the seating surface of the robot and the manipulator acting on the human body to assist the standing motion. At this time, the seating surface and the manipulator show different motions from the human body segments, and the ACT and the link do not need to be adjacent to the human body. Therefore, it is considered that the linear ACT is selected for the chair type because the torque is efficiently exerted by directly transmitting the force in the linear motion direction to the seating surface using the linear ACT, or by increasing the moment arm in the rigid type with the manipulator.

### 4.2.3. Designing of Control Strategy

The ACT is driven to exert a desired output of the exoskeleton or end-effector by a signal provided from the controller to assist in human operation. As for the control strategy of ACT, it was proven that position control tends to be selected for both the wearable type and the non-wearable type. In the wearable type, the input signal obtained from the angle sensor of the robot-driven joint is used for the start intention estimation of STS or walking motion. In wearable type, the joint angle change during STS and walking is reproduced in a predetermined trajectory by the angle control of the robot drive joint. In the wearable type in which such control is formed, it is often intended for support of a paralyzed person with spinal cord damage and cerebral vessel failure, because it is thought that robot based on the angle control is acting for the movement of lower limbs which the user performs.

On the other hand, there have been many studies in which position control of the non-wearable type aims to control the manipulator's posture and the user's center of gravity trajectory. The unworn type is intended to reproduce the user's natural STS, and the center of gravity trajectory is estimated from the input signals of the accelerometer and angle sensor installed in the user. There was a tendency for assistance with a high correlation with natural STS to be carried out by making the manipulator imitate the assistance technology of the specialist.

The force control was generally small for only three cases of the wearable type, and impedance control, which controls the interrelation between the force and the position, was seen in assistance systems of both the wearable type and the non-wearable type. In assisting with STS, many position controls have been adopted to control the user's joint angle and desired center of gravity trajectory; however, when aiming at drawing out the user's residual function, a design that can implement the STS assistance without much discomfort, such as by reducing the user's biological impedance. By detecting the joint torque generated in the user's lower limbs, the robot moves in cooperation with the joint angular velocity and angular displacement, and the discomfort felt by the wearer is alleviated.

### 4.2.4. System

Approximately half of the surveyed literature selected system designs by SISO. As a background for this, it is considered that human judgment and intuitive adjustment facilitate system design because the input–output relations are both single [48]. Therefore, it is thought that SISO is relatively more frequently reported for both the wearable type and the non-wearable type.

The reason why many SISO systems have been reported of the wearable type is because the cooperative movement of the hip and knee joints by the user is carried out when the voluntary movement of the person intervenes in the standing motion, so that the possibility of mutual interference between individual ACTs is low; therefore, it is considered that SISO controlled by one controller for one ACT is selected.

On the other hand, it is necessary to control all the driving joints simultaneously in the wearable type for people who have become paraplegic because of spinal cord damage and have remarkably impaired lower limb motor function. Additionally, since the non-wearable type assists the user in using a manipulator, it is necessary to control multiple ACTs simultaneously with the posture change of the manipulator. In such cases, since the ACT interaction for one manipulator may interfere with that of another, it is probable that MIMO will be selected because there is a need for the simultaneous control of two or more ACTs.

## 5. Conclusions

This review investigated STS assistance robots studied over the last 12 years. Wearable-type robots were more likely to use total assistance to help SCI patients because of the higher assist forces provided by the links, joints, and actuators placed parallel to the body. For these wearable-type robots, there was a tendency to select a motor that is lightweight and has a high output. On the other hand, many non-wearable type robots are of the chair type, which only aids the STS, and a leaning-on (walker) type, which assists both STS and walking. In both cases, the lower limb function left in the elderly body. The percentage of use of partial assistance was high. In control strategy, position control was often adopted mainly for both the wearable type and the non-wearable type. By controlling the joint angle of the manipulator, the user's natural center of gravity trajectory and the user's lower limb joint angle change were realized. In addition, both wearable type and non-wearable type actuators were controlled individually, and SISO was often found in the input/output relationship between the target amount and the sensor signal and control amount.

**Author Contributions:** K.N.: conceptualization and writing of the original draft; N.S.: discussion, review, and editing. All authors have read and agreed to the published version of the manuscript.

**Funding:** This research ware funded partially by Kwansei Gakuin University.

**Acknowledgments:** The authors acknowledge Kazufumi Tsukamoto for some of the preliminary literature collection for this review.

**Conflicts of Interest:** The authors declare no conflict of interest.

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
