# Peer review of "Current Status and Consideration of Support/Care Robots for Stand-Up Motion"

_applsci, doi:10.3390/app11041711_

Round 1
Reviewer 1 Report
-The author analyzed the previous studies on the Support/Care robot.
-However, the authors presented only the numbers for each classification after classifying the previous studies.
-In-depth analysis of the performance and utilization of previous studies is required.
-In addition, it is necessary to present pictures and performances of previous studies described as examples for authors.
-The English expression or grammar should be corrected, and the terms used are inconsistent, so this should be corrected.
Reviewer 2 Report
Dear authors,
the submitted article concerns the review of robots for assisting standing up motion. This is one of the first review article for such devices, what is a worthwhile for this article. However, as a review article, it has a very laconic introduction. We will not find any historical information about the beginnings of designing devices for the STS problem. Information on the clinical significance and effectiveness of such robots is lacking throughout the article. There was no evidence of the advantages of STS-only assistive devices compared to devices with a wide range of applications in gait training, such e.g. as those described in:
- Seo, K. H., & Lee, J. J. (2009). The development of two mobile gait rehabilitation systems. IEEE transactions on neural systems and rehabilitation engineering, 17(2), 156-166.
- Sabetian and J. M. Hollerbach, "A 3 wire body weight support system for a large treadmill," 2017 IEEE International Conference on Robotics and Automation (ICRA), Singapore, 2017, pp. 498-503, doi: 10.1109/ICRA.2017.7989062.
- Duda, S., Gąsiorek, D., Gembalczyk, G., Kciuk, S., & Mężyk, A. (2016). Mechatronic Device for Locomotor Training, Acta Mechanica et Automatica, 10(4), 310-315. doi: https://doi.org/10.1515/ama-2016-0049
- Frey, G. Colombo, M. Vaglio, R. Bucher, M. Jorg and R. Riener, "A Novel Mechatronic Body Weight Support System," in IEEE Transactions on Neural Systems and Rehabilitation Engineering, vol. 14, no. 3, pp. 311-321, Sept. 2006, doi: 10.1109/TNSRE.2006.881556.
The introduction should be more elaborate. The review method stated that articles from 2008-2018 were researched by google scholar. And by the IEEE Xplore database from 2008-2019. Why is 2019 omitted from Google? Has the PubMed database dedicated to medical issues been checked? In this database, for example, we can find an article from 2019 about A Semi-Wearable Robotic Device for Sit-to-Stand Assistance.
(H. Zheng, T. Shen, M. R. Afsar, I. Kang, A. J. Young and X. Shen, "A Semi-Wearable Robotic Device for Sit-to-Stand Assistance," 2019 IEEE 16th International Conference on Rehabilitation Robotics (ICORR), Toronto, ON, Canada, 2019, pp. 204-209, doi: 10.1109/ICORR.2019.8779425.)
The article contains many editing errors, e.g. the expression "system of control system" (line 50) or numbering "[29] .... [32]" instead of [29-32] (line 298). Sections 3.3 and 4.3 are called "Designing of control systems" but they are not about designing. It is rather a comparison of the used control systems, so a more appropriate subsection title would be simply "control systems".
References must be significantly improved. In Table 1, there are surnames in the "Author" column. The writing of the References section is inconsistent with the standards - sometimes we have a surname, sometimes first names, and sometimes the initials only as in 55 or 27. It is difficult to verify the correlation of these items with the table.
The summary seems to be incomplete. Is line 383 a continuation of the previous one? The summary should be completed. The article lacks any device from 2019 and newer. If the review mainly concerned the years up to 2018, maybe it is worth summarizing the progress made in recent years?
Round 2
Reviewer 1 Report
The following minor modifications are required.
- In Table 1 and Table 2, it would be good to indicate the reference number along with the first author.
- There are still many English expressions that are not well understood. I would appreciate it if you could correct it with a more readable English expression.
- In the classification of power units, controls, and systems, simply classifying them according to wearable and non-wearable does not give meaningful information to readers. In particular, in the case of non-wearable, it would be meaningful only when detailed classification is performed at the catagory level such as Chair, Sling, leaning-on and Handle.
Author Response
Revisions are indicated by yellow markers on the resubmitted papers.

Reviewer 2 Report
Dear authors,
in my opinion, the introduced changes increased the value and quality of the article. However, I have one more comment. Some subsections have been named "Designing of control". Please consider changing the title as their content does not concern the design of control systems. Rather, it is a description of a control strategy.
Author Response

(The authors gave the same response as above.)
